# Associations of Dietary Glycemic Index, Glycemic Load and Carbohydrate with the Risk of Cervical Intraepithelial Neoplasia and Cervical Cancer: A Case-Control Study

**DOI:** 10.3390/nu12123742

**Published:** 2020-12-04

**Authors:** Sundara Raj Sreeja, Sang Soo Seo, Mi Kyung Kim

**Affiliations:** 1Division of Cancer Epidemiology and Prevention, National Cancer Center, Ilsandong-gu, Goyang-si 10408, Korea; 1905209@ncc.re.kr; 2Center for Uterine Cancer, National Cancer Center, Ilsandong-gu, Goyang-si 10408, Korea; ssseomd@ncc.re.kr

**Keywords:** glycemic index, glycemic load, CIN, cervical cancer, carbohydrates

## Abstract

Background: The association of dietary glycemic index (GI) and glycemic load (GL) with the risk of cervical cancer has never been investigated. Thus, we aimed to find evidence of any association of GI and GL with the risk of cervical intraepithelial neoplasia (CIN) and cervical cancer. Methods: In this hospital-based case-control study, we included 1340 women (670 controls and 262, 187 and 221 patients with CIN1, CIN2/3, and cervical cancer, respectively) from the Korean human papillomavirus cohort study. Completed demographic questionnaires and semi-quantitative food-frequency questionnaires were collected. The association of dietary GI and GL with CIN and cervical cancer was estimated using a logistic regression model. Results: The multivariate odds ratios (OR) of the highest compared with the lowest quintile of GL for CIN1 were 2.8 (95% confidence interval (CI) = 1.33–5.88). Dietary GI and GL were not associated with CIN2/3 and cervical cancer. Stratified analyses by body mass index (BMI) indicated a positive association between GI and GL and CIN 1 risk among women with a BMI (in kg/m^2^) <23 (OR = 2.94; 95% CI = 1.32–6.53; *p* for trend = 0.031 for GI and OR = 3.15; 95% CI = 1.53–6.52; *p* for trend = 0.013 for GL), but not among those with a BMI of ≥23. A stratification analysis by menopausal status showed that the highest quintile of GI and GL was significantly associated with the risk of CIN1 (OR = 2.91; 95% CI = 1.43–5.96; *p* for trend = 0.005) (OR = 2.96; 95% CI = 1.53–5.69; *p* for trend = 0.023) among premenopausal women. Also, in HPV positive women, dietary GL showed significant CIN1 risk (OR = 2.61; 95% CI = 1.09–6.24; *p* for trend = 0.087). Conclusion: Our case-control study supports the hypothesized associations of dietary GI and GL with increased risk of CIN1. Thus, the consumption of low GI and GL foods plays a significant role in the prevention of cervical carcinogenesis.

## 1. Introduction

Dietary consumption of excessive glycemic index (GI) and glycemic load (GL) foods may play an inevitable part in the emergence of cancers [1]. Dietary GI indicates the blood glucose-raising potential of carbohydrate foods [2]. This index represents the increased area under the blood glucose response curve after a 50 g consumption of available carbohydrate from a test food relative to standard food (usually bread or glucose) [3]. GL represents the amount of carbohydrate in the food [4]. It is calculated by multiplying a given food GI value by the total carbohydrate content (g) and then dividing the product by 100 [5]. GI and GL are considered as important factors in clinical research exploring the effect of diet in disease etiology [6,7,8]. Foods with high GI may reduce insulin production and inflammatory responses [9].

GI and GL have been reported in coronary heart disease, diabetes mellitus, and obesity, but their associations with cancer are inconsistent [10]. Several epidemiological and meta-analysis studies have been performed on dietary GI and GL and their association with the risk of several cancers such as ovary [4,11], endometrium [12,13], breast [14,15], liver [16], colon [17] and pancreas [18]. However, these studies have reported conflicting results. Some studies found positive associations between dietary GI or GL and cancer risk [4,11,13,14,17], whereas others showed no association [12,13,16,18]. The African American Cancer Epidemiology Study of 406 cases and 609 controls found a significant association between dietary GI and GL and ovarian cancer risk in a follow-up period of 16 years [11]. In the prostate, lung, colorectal and ovarian cancer screening trial, high intake of carbohydrates and GL shows protective effect against endometrial cancer [13]. Similarly, in a European cohort study of 11,576 cases of breast cancer with a median follow-up of 11.5 years, ingestion of high carbohydrate and GI is positively associated with elevated risk of developing estrogen receptor-negative and estrogen receptor-negative/progesterone receptor-negative breast cancer among postmenopausal women [14].

Diets rich in carbohydrates cause insulin resistance resulting in chronic hyperinsulinemia [19]. High concentrations of insulin can reduce the formation of insulin-like growth factor binding protein-1 and develop the synthesis of insulin-like growth factor 1 (IGF-1), which can in turn increase the risk of cancers by inducing cell proliferation and differentiation, and inhibiting apoptosis and synthesis of sex steroids [20]. Several studies have reported that a high intake of carbohydrates results in elevated IGF levels, which may strengthen the emergence of cervical carcinogenesis, but others showed inconsistent and mixed findings [21,22,23,24,25]. IGF stimulates the mitogenic effect of epidermal growth factor on HT-3 cervical cancer cells, increasing cell proliferation [21]. Increased consumption of carbohydrate in the diet alters metabolism and increases the plasma level of inflammatory biomarkers and impairs glycogen synthesis, which leads to metabolic alterations [23]. By contrast, results from the Biomarkers of Cervical Cancer Risk study of 329 cases and 621 controls suggest that elevated levels of IGF-1 among younger women show a protective and shielding effect in the precursors of cervical cancer [24]. Moreover, consumption of complex carbohydrate and fiber obtained from whole grains may have a beneficial role in cancer prevention [25].

No previous epidemiological evidence of any association of dietary GI and GL with the risk of CINs and cervical cancer is available. One study reported that high blood glucose levels are significantly associated with an elevated risk of cervical cancer with poor prognosis [26]. The aim of the current study was to determine the association of dietary GI and GL with the risk of cervical intraepithelial neoplasia (CIN) and cervical cancer among Korean women.

## 2. Materials and Methods

### 2.1. Study Design and Population

The Korean human papillomavirus (HPV) cohort study has been conducted with approval from the Korean National Cancer Center’s Institutional Review Board (NCC2016-0147) since March 2006. The study subjects aged 18–65 years were screened from the gynecologic centers of eight university hospitals in Korea (Figure 1). The inclusion criteria were currently not pregnant, sexually active and no treatment for CINs or cervical cancer during 18 months prior to the enrollment. All subjects completed a structured lifestyle questionnaire and a semi-quantitative food frequency questionnaire (SQ-FFQ). They also underwent physical and gynecologic examinations including a hybrid capture II test and papanicolaou (Pap) smear. This study included 1495 subjects from the Korean HPV cohort study. Subjects with a history of cancer at other sites (*n* = 88) and with incomplete questionnaires (*n* = 67) were excluded. The participants were assigned into groups according to baseline Pap smear pathology: normal (*n* = 670), CIN1 (*n* = 262), CIN2/3 (*n* = 187) and cervical cancer (*n* = 221).

### 2.2. Assessment of Dietary Measurements

At the time of enrollment, questionnaires related to lifestyle and socio-demographic characteristics were collected. The lifestyle information included diverse medical data such as height, weight, history of pregnancy and oral contraceptive use, reproductive and menstrual status, and family history of diseases including cervical cancer. The socio-demographic characteristics such as marital status (single, married or separated), education level, physical status, smoking habits, and alcohol habits with detailed exposure times were also recorded. The subject’s medical records and pathology charts confirmed no history of CINs or cervical cancer. Subsequently, the pathological and laboratory information was entered into the epidemiological database.

For each subject, the usual dietary habits were recorded in the form of detailed intakes and standard portion sizes over the course of a year preceding enrollment using a 95-item SQ-FFQ instrument [27]. Food consumption frequency was divided into nine categories that ranged from “almost none” to “three times per day”. The standard portion size was classified into three categories “less than”, “one” or “more than” the advised size. The mean amount of usual food intake was computed by multiplying the frequency of consumption by the standard portion size for a single dish. The nutrient intake for each dish was computed using the Nutritional Analysis Program (version 4.0, Korea Nutrition Society, Seoul, Korea).

### 2.3. HPV Detection, Pap Smears and Histological Diagnosis

Cervical samples were collected using a Cervix-Brush (Rovers Medical Devices, Oss, The Netherlands). The Cervical-Brush was kept in a vial containing PreservCyt solution (Cytyc Corporation, Marlborough, MA, USA); the vial was placed into the Thin Prep Processor (Cytyc Corporation, Marlborough, MA, USA). Detection of oncogenic HPV was accomplished using hybrid capture II technology (HC-II; Digene Co., Silver Spring, MD, USA). HC-II analysis was performed to identify the HPV infection status of patients with the viral load (RLH/PC). Microplate chemiluminescence was measured for qualitative detection of all types of HPV in relative light units (RLUs). This test outcome was read as positive concentrations of 1 pg/mL or more than the RLU/cut off ratio (RLUs of specimen/mean RLUs of two positive controls). The cytological grade of Pap smears was classified according to the Bethesda Classification System [28]. The histological diagnosis was made based on the presence of cervical intraepithelial lesions.

### 2.4. Calculation of Dietary GI and GL

According to literature and international data, the GI values use glucose as a reference food (GI = 100). A value of 100 represents the standard amount of pure glucose [29]. In this study, GI values were assigned to each food item (*n* = 301), as follows. The GI values of common Korean foods were obtained based on the international tables of GI and GL values (step 1) [30]. We determined whether a common Korean food directly matched to a food in the international data (step 2), a common Korean food had a similar carbohydrate amount to a food in the international data (step 3), the cooking method was closely associated with that in the international data (step 4), and a variety of GI values existed for each food item in the international data (step 5). If there were little or no carbohydrates in foods, such as meat, fish, shellfish, eggs, certain vegetables and alcoholic liquors, a value of 0 was assigned (step 6). Dietary GI was calculated by multiplying the GI value of each food item by its carbohydrate content (g) and then dividing by the total amount of carbohydrate consumed (g/day). Dietary GL was computed by multiplying the GI value of an individual food item by its carbohydrate amount (g) and then dividing by 100.

### 2.5. Statistical Analysis

With regard to the distributions of continuous and categorical variables, ANOVA including post-hoc analyses using the Tukey test and Chi-squared (χ^2^) test were conducted in order to evaluate the differences among the normal, CINs, and cervical cancer groups. Multivariate logistic regression analysis was used to obtain the odds ratios (ORs) and corresponding 95% confidence intervals (95% CI) by quintiles of dietary GI and GL. Multivariate logistic regression model was adjusted for age (years), energy intake (kcal/day), marital status, education level, physical activity, smoking status, history of pregnancy, oral contraceptive use, hospitals and family history of cervical cancer, and which values were differentially distributed among the groups. Stratified analysis was conducted for body mass index (BMI), menopausal status and HPV status on the relationship of dietary GI and GL with CINs and cervical cancer risk. All statistical analyses were conducted using Statistical Analysis System software, version 9.4 (SAS Institute Inc., Cary, NC, USA). The reported *p* values were two-tailed and *p* values <0.05 were considered as statistically significant.

## 3. Results

The subject’s demographic characteristics, socioeconomic status and dietary intakes stratified by controls (*n* = 670), CIN1 (*n* = 262), CIN2/3 (*n* = 187) and cervical cancer (*n* = 221) are shown in Table 1. Patients with cervical cancer were older and more obese than the controls and patients with CIN1 and CIN2/3 (*p* < 0.0001). Significant differences in marriage, education, current drinker, pregnancy history, post menopause, and HPV infection were observed among the groups. Carbohydrate intake was not different among the groups, but GI and GL were different. Additionally, fruit intake was higher (*p* < 0.0001) in the control group than in the CIN1, CIN2/3, and cervical cancer groups. The CIN1 group showed a higher proportion of dietary intake values of energy (*p* = 0.0110), fat (*p* < 0.0001), protein (*p* = 0.0101), total fiber (*p* = 0.0002), white vegetables (*p* < 0.0001), total vegetables (*p* = 0.0029) and GI and GL (*p* < 0.0001).

The associations of dietary GI and GL with the risk of CIN1, CIN2/3 and cervical cancer are shown in Table 2. After adjusting for socioeconomic status and dietary intake, the multivariate ORs of the highest compared with the lowest quintile of dietary GL for CIN 1 were 2.8 (95% CI = 1.33–5.88). Dietary GI and GL were not significantly associated with the risk of CIN2/3 and cervical cancer. However, significant associations were obtained between dietary and GL on CIN1 risk (*p* = 0.005, *p* = 0.022), when they were expressed as continuous variables, but not with the risk of CIN2/3 and cervical cancer. Moreover, the association of carbohydrate intake and dietary GI and GL with the risk of CIN was not significant.

Table 3 shows the multivariate ORs of dietary GI and GL for the risk of CIN and cervical cancer stratified by BMI. Dietary GI and GL were significantly associated with the risk of CIN1 among women with BMI <23 (OR = 2.94; 95% CI = 1.32–6.53; *p* for trend = 0.031 for GI and OR = 3.15; 95% CI = 1.53–6.52; *p* for trend = 0.013 for GL) compared with the lowest and highest quintiles. For women with BMI ≥23, carbohydrate intake and dietary GI and GL were not associated with the risk of CIN and cervical cancer.

Multivariate ORs of dietary GI and GL for the risk of CIN and cervical cancer stratified by menopausal status are presented in Table 4. In premenopausal women, the highest quintile of dietary GI and GL, was significantly associated with the risk of CIN1 (OR = 2.91; 95% CI = 1.43–5.96; *p* for trend = 0.005) (OR = 2.96; 95% CI = 1.53–5.69; *p* for trend = 0.023). When dietary GI and GL were designated as continuous variables, increased risk of CINs and cervical was not observed among premenopausal women. Similarly, dietary GI and GL were not associated with the risk of CIN 1, CIN2/3 and cervical cancer among menopausal women.

Table 5 shows the multivariate ORs of dietary GI and GL (quintiles) for CINs and cervical cancer risk stratified by HPV infection status. Dietary GI had no significant association with the risk of CINs and cervical cancer in both the HPV negative and HPV positive groups. Among the HPV positive women, the highest quintile of dietary GL showed a significant association in CIN1 (OR = 2.61; 95% CI = 1.09–6.24; *p* trend = 0.087). Therefore, the results suggested a significant association between dietary GL and CIN1 risk among HPV positive women.

## 4. Discussion

To the best of our knowledge, this study is the first to investigate the association of dietary GI and GL with the risk of CINs and cervical cancer. We found that dietary GL was associated with the risk of CIN1 but not with the risk of CIN2/3 and cervical cancer. Stratification analysis indicated that the positive association of dietary GI and GL with the risk of CIN1 was predominant among women with a BMI <23 kg/m^2^, premenopausal and HPV positive.

The current study found that the consumption of GI and GL diet was statistically significantly associated with the risk of CIN1, but not with the risk of CIN2/3 and cervical cancer. A possible reason for this is that carbohydrate-rich foods inhibit hormone-binding globulin synthesis and stimulate cell proliferation [31]. Additionally, a dietary pattern with a high glycemic load and low intensity of LINE-1 methylation promotes epigenetic alterations and develops an increased risk of CIN [32]. Genetic alterations associated with insulin resistance contribute to genomic damage and chromosome instability, leading to precancerous lesions [26]. In an American study, a high carbohydrate diet was associated with elevated levels of IGF-1 and IGFBP-3 levels among premenopausal women, which upregulated the progesterone receptor and facilitated cell proliferation [33]. In our previous studies, HPV infection with cofactors such as alcohol consumption, viral load and smoking was associated only with CIN1 risk [34,35]. Even though it is difficult to explain how GI and GL diet contribute to CIN development, these factors can be critical compared to other known factors. However, most of the studies related to dietary patterns for the risk of cervical cancer were focused largely on folic acid, carotenoids, and vitamin A [36,37,38]. Thus, further research is necessary to analyze the effect of carbohydrate diet, and dietary GI and GL on the risk of CIN and cervical cancer.

This study showed a significant association between CIN1 risk and dietary GL, but not GI. Similarly, high GL (but not GI diets) substantially increased gastric cancer risk [39]. High glycemic foods increase the hunger compared to low-glycemic foods, which in turn impairs glucose metabolism [40]. In a cross-sectional study conducted in Italy, consumption of a western diet rich in carbohydrate content, such as starch, sucrose, and lactose, was significantly associated with high incidence of cervical cancer [41]. Consumption of legumes and complex carbohydrates shows a higher risk for cervical cancer than plant-based nutrition rich in fiber and phytonutrients [42]. The lack of association between GI and CIN risk suggests that the quantity and quality of carbohydrates intake may be related to cervical lesions.

Our study showed that dietary GI and GL were strongly associated with the risk of CIN in women with a normal BMI range of <23 kg/m^2^. Similarly, in a case-control study among a Korean population involving 582 cases and 543 controls, a positive association was found between obesity and the risk of CIN by the increased intake of carbohydrate diet, which in turn affects estrogen metabolism [43]. Moreover, when compared with women with a normal BMI, obese and overweight women were associated with a two-fold elevated risk of developing cervical carcinoma by a diet rich in carbohydrates, fats, and sugars [44]. Thus, BMI may be considered as an effect modifier in the association of diet with the risk of cervical cancer. Further studies are warranted to explore the relationship between BMI and individual foods.

In our study, a significant association was found between dietary GI and GL and with the risk of CIN among premenopausal women. By contrast, postmenopausal women showed an increased risk of developing CIN by the intake of high GL foods such as white bread, white rice and potatoes, which results in an alteration in hormone levels [38]. The incidence of cervical cancer is elevated among postmenopausal women by high carbohydrate diet, and menopausal status is considered as a significant risk for cervical carcinogenesis [45]. Further investigations need to be conducted to explore the association between dietary GI and GL and the risk of CIN and cervical cancer stratified by menopausal status.

Human papilloma virus (HPV), the most common viral infection in reproductive organs, is a major etiologic factor for cervical intraepithelial neoplasia and cervical cancer [46]. The primary risk factor for CINs and cervical cancer accounts for HPV 16 and 18 oncogenic genotypes [47]. HPV infection alone, however, might not be a sufficient cause for cervical carcinogenesis, because one of the plausible risk factors for its development is diet [41]. Our results showed that HPV-positive women with a high GL diet were at a high CIN1 risk. Similarly, in a Brazilian study, cervical cancer patients had diets with high glycemic loads [42]. Moreover, elevated blood glucose levels were recorded in CIN and cervical cancer cases and was considered as a risk factor for the development of cervical lesions [48].

High glycemic foods break down very quickly and cause sudden fluctuations in blood sugar level [49]. They also increase the production of insulin growth factor which contributes to the development of cervical lesions [50]. Insufficient consumption of plant-based foods and dietary nutrients, such as vitamins, minerals, and carotenoids, stimulates the expression of viral oncogenes E6 and E7, thereby increasing the risk of CIN [51]. Diets with low GI are considered as cancer prevention dietary nutrients, which are slowly digested and maintain a stable blood glucose level [1]. Increased intake of high GI foods followed by less fruit consumption and folate deficiency reduces the breakdown of complex carbohydrates and leads to LINE-1 hypomethylation in women, which further contributes to the development of cancer [52]. Thus, consumption of a low glycemic diet and healthy lifestyle habits reduce the risk of cervical cancer.

Korean diets involve a high intake of carbohydrate rich foods such as cereal grains, bread, noodles and pasta, and starchy vegetables [53]. Japanese diets have high GI and GL values of 65–67 and 141–185, respectively [54]. Additionally, white rice in Korean diets have high GI values of 51–93, while in the Italian diet, the GI value is 64 and 69 in the international GI databases [55]. These inconsistent results across nations in dietary GI and GL may be due to the diverse nature of dietary patterns, dietary assessment methods and dietary habits. Further studies are required to explore the carbohydrate content and differences in the quality of dietary patterns.

Our study includes several strengths. The extensive data collection in the study allowed us to address potential confounding by anthropometric and lifestyle factors. To the best of our understanding, this study is the first to investigate the association of dietary GI and GL with the risk of cervical dysplasia. Another advantage of the current study was its adjustment of energy consumption over the preceding year to control for potential confounding effects, because carbohydrates are a major source of energy. Moreover, this study assessed the association of dietary GI and GL for both cervical cancer and CIN cases based on dietary GI and GL and carbohydrate intake. Nevertheless, this study has several limitations. Firstly, as a case-control design, recall and selection biases are possible. Secondly, this study focused only on a Korean population, and the results cannot be applied to the general population because dietary habits may vary among various ethnic groups. Thirdly, misreporting or falsification in the data of dietary intake can complicate the association between diet and health outcomes. Additionally, our study used the SQ-FFQ, which was not designed specifically for the calculation of dietary GI and GL. HPV status data were not obtained for all subjects, and therefore missing data could also influence the results. Finally, the present study included only a restricted number of dishes or foods and lacked information on the food consumption of dishes.

## 5. Conclusions

Our case-control study supports the hypothesized associations of dietary GI and GL with increased risk of CIN 1. Additional studies including larger sample sizes and molecular approaches are necessary to identify those associations and reveal their underlying mechanisms. Given the potential associations of dietary GI and GL with increased risk of precancerous cervical dysplasia, our findings may have important implications for reducing cervical intraepithelial neoplasia, and thereby preventing the progression to cervical cancer.

## Figures and Tables

**Figure 1 nutrients-12-03742-f001:**
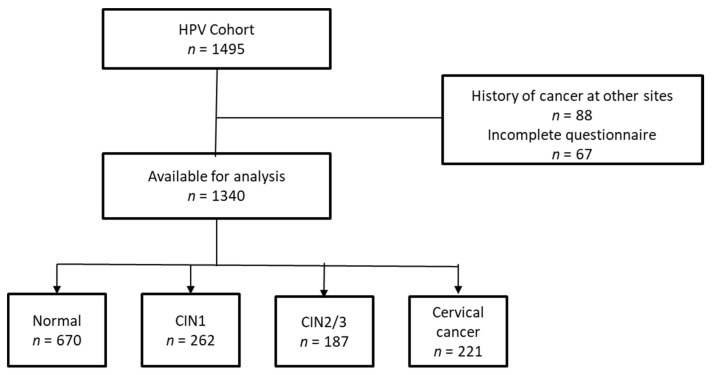
Flow diagram of included study subjects. (HPV, human papillomavirus; CIN, cervical intraepithelial neoplasia).

**Table 1 nutrients-12-03742-t001:** General characteristic of study subjects.

Variables	Normal (*n* = 670)	CIN1 (*n* = 262)	CIN2/3 (*n* = 187)	Cervical Cancer (*n* = 221)	*p* Value ^1^
Age (years), mean ± SD	43.6 ± 10.6 ^b^	38.7 ± 11.1 ^c^	39.9 ± 10.5 ^c^	50.2 ± 12.2 ^a^	<0.0001
BMI (kg/m^2^), mean ± SD	22.4 ± 2.9 ^b^	21.8 ± 2.8 ^c^	21.9 ± 3.2 ^bc^	23.5 ± 3.6 ^a^	<0.0001
≥23, *n* (%)	251 (37.5)	79 (30.3)	57 (30.7)	112 (50.7)	<0.0001
Marriage, *n* (%)	528 (78.8)	174 (66.4)	135 (72.2)	153 (69.2)	<0.0001
Education ≥ College, *n* (%)	236 (35.3)	107 (40.8)	52 (28.0)	25 (11.3)	<0.0001
Physical activity (MET-h/d), mean ± SD	37.1 ± 7.5	37.7 ± 8.5	36.0 ± 6.6	37.8 ± 9.6	0.068
Active physical activity, *n* (%) ^2^	171 (25.5)	68 (26.0)	33 (17.7)	42 (19.0)	0.037
Smoking status, *n* (%)					
Smoker	50 (7.5)	32 (12.2)	23 (12.4)	20 (9.1)	0.135
Past smoker	32 (4.8)	17 (6.5)	12 (6.5)	14 (6.3)	
Alcohol intake, *n* (%)					
Drinker	370 (55.2)	190 (72.5)	114 (61.0)	87 (39.4)	<0.0001
Past drinker	24 (3.6)	7 (2.7)	17 (9.1)	28 (12.7)	
History of pregnancy, *n* (%)	557 (83.3)	183 (70.1)	144 (77.0)	206 (94.1)	<0.0001
Oral contraceptive use, *n* (%) ^3^	107 (16.0)	46 (17.6)	44 (23.7)	44 (20.0)	0.091
Post-menopause, *n* (%)	228 (34.1)	47 (18.0)	37 (19.8)	139 (63.2)	<0.0001
HPV, *n* (%) ^4^					
Positive	208 (45.5)	117 (70.1)	51 (63.8)	31 (56.4)	<0.0001
Negative	249 (54.5)	50 (29.9)	29 (36.3)	24 (43.6)	
Family history of cervical cancer, *n* (%)	23 (3.4)	14 (5.3)	9 (4.8)	12 (5.4)	0.440
Dietary intake, mean ± SD					
Energy (kcal/day)	1892 ± 514 ^ab^	1991 ± 603 ^a^	1889 ± 508 ^ab^	1833 ± 547 ^b^	0.011
Carbohydrate (g/day)	304 ± 80.6	312 ± 90.4	303 ± 81.3	303 ± 81.9	0.460
Fat (g/day)	41.4 ± 19.4 ^a^	44.6 ± 21.4 ^a^	40.0 ± 17.4 ^ab^	36.3 ± 18.4 ^b^	<0.0001
Protein (g/day)	70.8 ± 24.4 ^ab^	74.7 ± 28.1 ^a^	68.8 ± 23.4 ^ab^	67.3 ± 26.9 ^b^	0.010
Folate (μg/day)	393 ± 166	406 ± 176	381 ± 183	383 ± 192	0.401
Total fiber (g/day)	9.1 ± 3.9 ^a^	9.5 ± 4.3 ^a^	7.8 ± 3.3 ^b^	8.8 ± 4.5 ^ab^	0.0002
Cereals (g/day)	227 ± 78.2 ^b^	225 ± 85.2 ^ab^	234 ± 87.2 ^ab^	243 ± 76.8 ^a^	0.032
Total vegetables (g/day) ^5^	327 ± 151 ^ab^	349 ± 174 ^a^	296 ± 140 ^b^	341 ± 213 ^a^	0.003
Green and yellow vegetables (g/day)	121 ± 64.4	122 ± 72.2	124 ± 66.2	134 ± 87.0	0.239
White vegetables (g/day)	207 ± 108 ^a^	228 ± 121 ^a^	173 ± 87.3 ^b^	207 ± 140 ^a^	<0.0001
Fruits (g/day)	271 ± 334 ^a^	269 ± 285 ^a^	198 ± 200 ^b^	182 ± 184 ^b^	<0.0001
Dietary glycemic index (GI) ^6^	61.8 ± 20.4 ^b^	66.2 ± 23.0 ^a^	64.2 ± 23.4 ^ab^	57.0 ± 20.4 ^c^	<0.0001
Dietary glycemic load (GL) ^6^	177 ± 42.7 ^b^	192 ± 41.4 ^a^	182 ± 41.3 ^b^	163 ± 43.2 ^c^	<0.0001

^1^ ANOVA and Chi-squared test including post-hoc analyses using the Tukey method were used to assess the significant differences in subject distributions in continuous and categorical variables, respectively. ^2^ Physical activity was classified as sufficiently active (physical activity practiced for at least 30 min daily, 5 days per week, at moderate intensity or for at least 20 min daily, 3 days per week, at vigorous intensity) or insufficiently active. ^3^ Oral contraceptive use includes current and past use. ^4^ HPV was detected using the hybrid capture II assay. ^5^ Green and yellow vegetables and white vegetables are included in the “total vegetables”. ^6^ Dietary glycemic index and load were calculated based on glucose (=100 scale). ^a,b,c,ab,bc^ Mean and standard deviation in each group. BMI, body mass index; HPV, human papillomavirus

**Table 2 nutrients-12-03742-t002:** Multivariate ORs of dietary glycemic index (GI) and glycemic load (GL) for the risk of CIN and cervical cancer.

Variables	Normal	CIN1		CIN2/3		Cervical Cancer	
	*n* (%)	*n* (%)	OR (95% CI) ^1^	*n* (%)	OR (95% CI) ^1^	*n* (%)	OR (95% CI) ^1^
Dietary GI ^2^							
Q1 (<44.98)	135 (20.15)	37 (14.1)	1 (ref)	33 (17.7)	1 (ref.)	60 (27.2)	1 (ref.)
Q2 (44.98–54.45)	133 (19.85)	47 (17.9)	1.21 (0.65–2.24)	38 (20.3)	0.53 (0.21–1.33)	59 (26.7)	1.13 (0.45–2.84)
Q3 (54.45–63.94)	135 (20.2)	59 (22.5)	1.52 (0.81–2.84)	39 (20.9)	0.56 (0.23–1.35)	45 (20.4)	0.91 (0.33–2.52)
Q4 (63.94–77.24)	134 (20.0)	53 (20.2)	1.39 (0.71–2.71)	33 (17.7)	0.55 (0.22–1.39)	24 (10.9)	0.60 (0.19–1.89)
Q5 (>77.24)	133 (19.85)	66 (25.2)	1.34 (0.62–2.88)	44 (23.5)	1.0 (0.37–2.68)	33 (14.9)	0.74 (0.21–2.66)
*p* for linear trend ^3^			0.767		0.332		0.843
*p* for continuous ^4^			0.236		0.37		0.369
Dietary GL ^2^							
Q1 (<138.79)	134 (20.0)	28 (10.7)	1 (ref.)	31 (16.6)	1 (ref.)	64 (29.0)	1 (ref.)
Q2 (138.79–167.12)	134 (20.0)	45 (17.2)	1.76 (0.86–3.60)	39 (20.9)	1.27 (0.48–3.35)	58 (26.2)	1.36 (0.56–3.33)
Q3 (167.12–190.48)	134 (20.0)	45 (17.2)	1.36 (0.64–2.86)	35 (18.7)	0.75 (0.27–2.01)	33 (14.9)	0.70 (0.25–1.96)
Q4 (190.48–214.78)	135 (20.1)	53 (20.2)	1.02 (0.47–2.21)	38 (20.3)	1.15 (0.46–2.90)	38 (17.2)	0.55 (0.17–1.84)
Q5 (>214.78)	133 (19.9)	91 (34.7)	2.8 (1.33–5.88)	44 (23.5)	1.35 (0.74–2.48)	28 (12.7)	0.67 (0.20–2.33)
*p* for linear trend ^3^			0.005		0.716		0.468
*p* for continuous ^4^			0.022		0.507		0.093
Carbohydrate ^2^							
Q1 (<232.51)	134 (20.0)	49(18.7)	1 (ref.)	34(18.1)	1 (ref.)	43(19.4)	1 (ref.)
Q2 (232.51–279.64)	134 (20.0)	48(18.3)	1.35 (0.66–2.75)	39(20.9)	1.61 (0.64–4.06)	36(16.3)	0.92 (0.27–3.18)
Q3 (279.64–321.83)	135 (20.2)	49(18.7)	1.45 (0.67–3.10)	38(20.3)	1.31 (0.48–3.60)	62(28.1)	1.31 (0.34–5.01)
Q4 (321.83–362.87)	133 (19.9)	46(17.6)	1.60 (0.70–3.63)	37(19.8)	1.59 (0.48–5.21)	36(16.3)	1.10 (0.24–5.11)
Q5 (>362.87)	134 (20.0)	70(26.7)	2.24 (0.81–6.21)	39(20.9)	3.12 (0.69–14.22)	44(19.9)	0.34 (0.04–3.06)
*p* for linear trend ^3^			0.651		0.546		0.366
*p* for continuous ^4^			0.414		0.413		0.204

^1^ Adjusted for age (years) and energy intake (kcal/day) as continuous variables and for marital status (single, married, widowed/divorced/separated), educational status (≤high school graduate, ≥college), physical activity (sufficient, insufficient), smoking status (non-smoker, past-smoker, smoker), history of pregnancy, oral contraceptive use (non-user, past-/current-), and family history of cervical cancer, and hospitals (H01, H03, Ho4, H05, H06, H07, H12, NCC) as categorical variables. ^2^ Dietary GI and GL, and carbohydrate were each divided into five groups based on the normal group. Q1 and Q5 were the lowest and highest quintile groups, respectively. ^3^ P value was calculated for the linear trend of multivariate odds ratio. ^4^
*p* value was calculated by taking dietary GI and GL as continuous variables. Ref, standard ie. if Q1 is 1, others increases risk on odds ratio. OR, odds ratio; CIN, cervical intraepithelial neoplasia; GI, glycemic index; GL, glycemic load.

**Table 3 nutrients-12-03742-t003:** Multivariate ORs of dietary glycemic index (GI) and glycemic load (GL) for the risk of CIN and cervical cancer stratified by BMI.

Variables	Normal	CIN1		CIN2/3		Cervical Cancer	
	*n* (%)	*n* (%)	OR (95% CI) ^1^	*n* (%)	OR (95% CI) ^1^	*n* (%)	OR (95% CI) ^1^
BMI < 23 Dietary GI ^2^							
Q1 (<46.56)	83(19.8)	20(11.0)	1 (ref.)	24(18.6)	1 (ref.)	37(33.9)	1 (ref.)
Q2 (46.56–55.85)	84(20.1)	29(15.9)	1.33 (0.66–2.70)	25(19.4)	0.98 (0.46–2.08)	20(18.4)	0.80 (0.36–1.81)
Q3 (55.85–65.94)	84(20.1)	50(27.5)	2.47 (1.24–4.94)	23(17.8)	0.65 (0.30–1.39)	19(17.4)	0.66 (0.28–1.56)
Q4 (65.94–79.99)	84(20.1)	32(17.6)	1.62 (0.77–3.42)	28(21.7)	0.63 (0.29–1.37)	16(14.7)	0.54 (0.22–1.34)
Q5 (>79.99)	83(19.9)	51(28.0)	2.94 (1.32–6.53)	29(22.5)	0.79 (0.34–1.85)	17(15.6)	0.46 (0.17–1.21)
*p* for linear trend ^3^			0.031		0.644		0.565
*p* for continuous ^4^			0.107		0.202		0.732
Dietary GL ^2^							
Q1 (<143.28)	84(20.1)	17(9.3)	1 (ref.)	24(18.6)	1 (ref.)	37(33.9)	1 (ref.)
Q2 (143.28–171.71)	84(20.1)	24(13.2)	1.29 (0.62–2.71)	24(18.6)	1.21 (0.54–2.32)	27(24.8)	0.79 (0.37–1.70)
Q3 (171.71–195.27)	84(20.1)	33(18.1)	1.58 (0.77–3.26)	16(12.4)	0.64 (0.29–1.42)	14(12.8)	0.49 (0.20–1.21)
Q4 (195.27–217.91)	84(20.1)	36(19.8)	1.81 (0.88–3.74)	29(22.5)	0.99 (0.48–2.05)	20(18.4)	0.87 (0.37–2.02)
Q5 (>217.91)	82(19.62)	72(39.6)	3.15 (1.53–6.52)	36(27.9)	1.13 (0.53–2.43)	11(10.1)	0.50 (0.19–1.30)
*p* for linear trend ^3^			0.013		0.600		0.454
*p* for continuous ^4^			0.018		0.391		0.566
Carbohydrate ^2^							
Q1 (<227.93)	83(19.86)	38(20.9)	1 (ref.)	24(18.6)	1 (ref.)	20(18.4)	1 (ref.)
Q2 (227.93–277.66)	86(20.57)	35(19.2)	0.89 (0.47–1.68)	26(20.1)	1.15 (0.53–2.48)	18(16.5)	1.27 (0.48–3.34)
Q3 (277.66–316.17)	83(19.9)	27(14.9)	0.79 (0.38–1.64)	23(17.8)	1.17 (0.49–2.81)	28(25.7)	2.33 (0.78–6.94)
Q4 (316.17.83–361.78)	84(20.1)	35(19.2)	0.94 (0.43–2.02)	28(21.7)	1.66 (0.62–4.43)	18(16.5)	2.11 (0.58–7.74)
Q5 (>361.78)	82(19.62)	47(25.8)	1.01 (0.37–2.74)	28(21.7)	2.41 (0.68 − 8.64)	25(22.9)	2.19 (0.43 − 11.16)
*p* for linear trend ^3^			0.941		0.606		0.575
*p* for continuous ^4^			0.998		0.982		0.720
BMI ≥ 23 Dietary GI ^2^							
Q1 (<43.32)	51(20.32)	18(22.8)	1 (ref.)	8(14.0)	1 (ref.)	20(17.9)	1 (ref.)
Q2 (43.32–51.69)	50(19.9)	15(19.0)	0.80 (0.34–1.88)	8(14.0)	0.61 (0.19–1.94)	32(28.6)	1.18 (0.53–2.62)
Q3 (51.69–59.84)	51(20.4)	14(17.7)	0.84 (0.35–2.04)	20(35.1)	1.81 (0.67–4.93)	24(21.4)	1.15 (0.50–2.65)
Q4 (59.84–73.02)	49(19.5)	19(24.1)	1.43 (0.60–3.14)	10(17.6)	0.78(0.26–2.40)	23(20.5)	1.08 (0.44–2.61)
Q5 (>73.02)	50(19.9)	13(16.5)	0.84 (0.30–2.34)	11(19.3)	0.90 (0.28–2.94)	13(11.6)	0.72 (0.26–1.99)
*p* for linear trend ^3^			0.663		0.211		0.876
*p* for continuous ^4^			0.461		0.318		0.393
Dietary GL ^2^							
Q1 (<130.87)	50(19.9)	11(13.9)	1 (ref.)	4(7.02)	1 (ref.)	27(24.1)	1 (ref.)
Q2 (130.87–161.40)	50(19.9)	21(26.6)	2.09 (0.84–5.18)	17(29.8)	4.31 (1.25–14.88)	29(25.9)	1.14 (0.53–2.46)
Q3 (161.40–183.08)	51(20.3)	12(15.2)	1.10 (0.41–2.94)	20(35.1)	4.56 (1.34–15.57)	24(21.4)	1.06 (0.47–2.38)
Q4 (183.08–207.80)	51(20.3)	18(22.8)	1.30 (0.51–3.32)	8(14.0)	1.85 (0.48–7.20)	14(12.5)	0.45 (0.18–1.15)
Q5 (>207.80)	49(19.5)	17(21.5)	1.70 (0.62–4.65)	8(14.0)	2.25 (0.55–9.26)	18(16.1)	0.73 (0.28–1.90)
*p* for linear trend ^3^			0.563		0.055		0.320
*p* for continuous ^4^			0.611		0.571		0.070
Carbohydrate ^2^							
Q1 (<232.5)	50(19.9)	11(13.9)	1 (ref.)	9(15.8)	1 (ref.)	22(19.6)	1 (ref.)
Q2 (232.5–289.13)	51(20.3)	15(19.0)	2.21 (0.74–6.58)	14(24.6)	2.24 (0.69–7.30)	22(19.6)	0.74 (0.29–1.90)
Q3 (289.13–331.78)	50(19.9)	19(24.1)	2.86 (0.89–9.20)	17(29.8)	3.65 (0.96–13.92)	37(33.1)	1.02 (0.36–2.93)
Q4 (331.78–367.01)	51(20.3)	11(13.9)	1.39 (0.36–5.37)	9(15.8)	1.21 (0.21–6.45)	12(10.7)	0.25 (0.06–0.94)
Q5 (>367.01)	49(19.5)	23(29.1)	4.10 (0.74–22.79)	8(14.0)	3.42 (0.40–29.61)	19(17.0)	0.36 (0.06–2.31)
*p* for linear trend ^3^			0.185		0.110		0.062
*p* for continuous ^4^			0.114		0.095		0.091

^1^ Adjusted for age (years) and energy intake (kcal/day) as continuous variables and for marital status (single, married, widowed/divorced/separated), educational status (≤ high school graduate, ≥college), physical activity (sufficient, insufficient), smoking status (non-smoker, past-smoker, smoker), history of pregnancy, oral contraceptive use (non-user, past-/current-) and family history of cervical cancer, and hospitals (H01, H03, Ho4, H05, H06, H07, H12, NCC) as categorical variables. ^2^ Dietary GI and GL and carbohydrate were each divided into five groups based on the normal group. Q1 and Q5 were the lowest and highest quintile groups, respectively. ^3^
*p* value was calculated for the linear trend of multivariate odds ratio. ^4^
*p* value was calculated by taking dietary GI and GL as continuous variables. Ref, standard ie. if Q1 is 1, others increases risk on odds ratio. OR, odds ratio; CIN, cervical intraepithelial neoplasia; GI, glycemic index; GL, glycemic load; BMI, body mass index.

**Table 4 nutrients-12-03742-t004:** Multivariate ORs of dietary glycemic index (GI) and glycemic load (GL) for the risk of CIN and cervical cancer stratified by menopausal status.

Variables	Normal	CIN1		CIN2/3		Cervical Cancer	
	*n* (%)	*n* (%)	OR (95% CI) ^1^	*n* (%)	OR (95% CI) ^1^	*n* (%)	OR (95% CI) ^1^
No Dietary GI ^2^							
Q1 (<47.03)	90(20.41)	29(13.6)	1 (ref.)	28(18.7)	1 (ref.)	26(32.1)	1 (ref.)
Q2 (47.03–56.86)	87(19.73)	40(18.7)	1.38 (0.75–2.53)	29(19.3)	0.90 (0.45–1.79)	18(22.2)	0.78 (0.34–1.78)
Q3 (56.86–66.18)	87(19.73)	57(26.6)	2.39 (1.30–4.41)	32(21.3)	1.02 (0.51–2.02)	11(13.6)	0.57 (0.22–1.47)
Q4 (66.18–79.34)	92(20.86)	31(14.5)	1.25 (0.64–2.43)	27(18.0)	0.62 (0.30–1.28)	17(21.0)	0.55 (0.22–1.33)
Q5 (>79.34)	85(19.27)	57(26.6)	2.91 (1.43–5.96)	34(22.7)	1.21 (0.55–2.67)	9(11.11)	0.48 (0.16–1.43)
*p* for linear trend ^3^			0.005		0.381		0.605
*p* for continuous ^4^			0.281		0.307		0.824
Dietary GL ^2^							
Q1 (<147.48)	88(19.95)	21(9.8)	1 (ref.)	26(17.3)	1 (ref.)	22(27.2)	1 (ref.)
Q2 (147.48–175.68)	88(19.95)	39(18.2)	1.77 (0.92–3.38)	34(22.7)	1.79 (0.92–3.47)	20(24.7)	1.02 (0.46–2.27)
Q3 (175.68–198.55)	89(20.2)	38(17.8)	1.61 (0.84v 3.1)	25(16.7)	1.22 (0.61–2.46)	12(14.8)	0.78 (0.31–1.95)
Q4 (198.55–219.80)	88(19.95)	42(19.6)	1.92 (0.99–3.71)	26(17.3)	1.17 (0.58–2.36)	17(21.0)	1.09 (0.45–2.62)
Q5 (>219.80)	88(19.95)	74(34.6)	2.96 (1.53–5.69)	39(26.0)	1.73 (0.85–3.54)	10(12.3)	0.70 (0.25–1.97)
*p* for linear trend ^3^			0.023		0.345		0.901
*p* for continuous ^4^			0.074		0.469		0.522
Carbohydrate ^2^							
Q1 (<232.67)	89(20.18)	39(18.2)	1 (ref.)	26(17.3)	1 (ref.)	12(14.8)	1 (ref.)
Q2 (232.67–281.75)	87(19.73)	44(20.5)	1.15 (0.64–2.09)	36(24.0)	1.47 (0.72–3.00)	16(19.8)	1.26 (0.46–3.47)
Q3 (281.75–324.31)	90(20.41)	37(17.3)	1.01 (0.51–1.99)	31(20.7)	1.21 (0.54–2.71)	24(29.6)	1.91 (0.63–5.81)
Q4 (324.31–371.11)	88(19.95)	38(17.8)	0.91 (0.43–1.89)	33(22.0)	1.16 (0.46–2.92)	13(16.0)	1.13 (0.30–4.31)
Q5 (>371.11)	87(19.73)	56(26.2)	1.26 (0.48–3.30)	24(16.0)	1.54 (0.44–5.47)	16(19.8)	1.16 (0.19–7.10)
*p* for linear trend ^3^			0.827		0.815		0.612
*p* for continuous ^4^			0.966		0.547		0.486
Yes							
Dietary GI ^2^							
Q1 (<42.08)	45(19.7)	11(23.4)	1 (ref.)	5(13.5)	1 (ref.)	31(22.3)	1 (ref.)
Q2 (42.08–49.74)	46(20.2)	8(17.0)	0.83 (0.27–2.54)	8(21.6)	1.61 (0.41–6.34)	26(18.7)	0.79 (0.35–1.77)
Q3 (49.74–59.25)	46(20.2)	8(17.0)	1.08 (0.35–3.32)	11(29.8)	1.99 (0.52–7.54)	37(26.6)	1.07 (0.49–2.32)
Q4 (59.25–74.06)	47(20.6)	15(31.9)	1.79 (0.64–5.00)	8(21.6)	1.18 (0.30–4.69)	25(18.0)	0.64 (0.28–1.46)
Q5 (>74.06)	44(19.3)	5(10.7)	0.49 (0.12–1.97)	5(13.5)	0.51 (0.10–2.53)	20(14.4)	0.57 (0.22–1.48)
*p* for linear trend ^3^			0.334		0.460		0.600
*p* for continuous ^4^			0.680		0.423		0.201
Dietary GL ^2^							
Q1 (<125.81)	45(19.7)	9(19.2)	1 (ref.)	7(18.9)	1 (ref.)	33(23.7)	1 (ref.)
Q2 (125.81–151.2)	47(20.6)	12(25.5)	1.41 (0.48–4.16)	8(21.6)	1.24 (0.35–4.39)	36(25.9)	1.20 (0.57–2.53)
Q3 (151.2–176.53)	45(19.7)	5(10.6)	0.81 (0.22–2.94)	12(32.5)	1.61 (0.49–5.28)	31(22.3)	0.84 (0.38–1.85)
Q4 (176.53–202.47)	47(20.6)	10(21.3)	1.29 (0.42–4.02)	6(16.2)	0.76 (0.20–2.93)	19(13.7)	0.39 (0.15–0.96)
Q5 (>202.47)	44(19.3)	11(23.4)	1.47 (0.47–4.64)	4(10.8)	0.53 (0.12–2.42)	20(14.4)	0.5 (0.20–1.25)
*p* for linear trend ^3^			0.870		0.567		0.100
*p* for continuous ^4^			0.205		0.568		0.064
Carbohydrate ^2^							
Q1 (<230.58)	45(19.7)	9(19.2)	1 (ref.)	8(21.6)	1 (ref.)	28(20.1)	1 (ref.)

Q2 (230.58–276.66)	46(20.2)	4(8.5)	0.74 (0.16–3.39)	4(10.8)	0.80 (0.17–3.89)	22(15.8)	1.02 (0.41–2.53)
Q3 (276.66–320.37)	47(20.6)	14(29.8)	2.57 (0.63–10.49)	11(29.8)	4.03 (0.71–22.73)	39(28.1)	1.88 (0.68–5.18)
Q4 (320.37–354.99)	46(20.2)	10(21.25)	2.50 (0.50–12.50)	9(24.3)	3.25 (0.42–25.11)	22(15.8)	1.18 (0.35–3.92)
Q5 (>354.99)	45(19.7)	10(21.28)	2.32 (0.29–18.8)	5(13.5)	3.26 (0.21–50.26)	28(20.1)	0.90 (0.19–4.22)
*p* for linear trend ^3^			0.396		0.275		0.399
*p* for continuous ^4^			0.060		0.839		0.549

^1^ Adjusted for age (years) and energy intake (kcal/day) as continuous variables and for marital status (single, married, widowed/divorced/separated), educational status (≤high school graduate, ≥college), physical activity (sufficient, insufficient), smoking status (non-smoker, past-smoker, smoker), history of pregnancy, oral contraceptive use (non-user, past-/current-), and family history of cervical cancer, and hospitals (H01, H03, Ho4, H05, H06, H07, H12, NCC) as categorical variables. ^2^ Dietary GI and GL and carbohydrate were each divided into five groups based on the normal group. Q1 and Q5 were the lowest and highest quintile groups, respectively. ^3^
*p* value was calculated for the linear trend of multivariate odds ratio.^4^
*p* value was calculated by taking dietary GI and GL as continuous variables. Ref, standard ie. if Q1 is 1, others increases risk on odds ratio. OR, odds ratio; CIN, cervical intraepithelial neoplasia; GI, glycemic index; GL, glycemic load.

**Table 5 nutrients-12-03742-t005:** Multivariate ORs of dietary GI and GL for CINs and cervical cancer risk with subjects stratified into HPV-negative and positive groups ^1^.

Variables	HPV-Negative ^2^			HPV-Positive ^2^		
	Normal	CIN1		Normal	CIN1	
	*n* (%) ^3^	*n* (%) ^3^	OR (95% CI) ^4^	*n* (%) ^3^	*n* (%) ^3^	OR (95% CI) ^4^
Dietary GI ^5^						
Q1(<44.14)	49(19.68)	8(16)	1 (ref.)	42(20.19)	26(22.22)	1 (ref.)
Q2(44.14–53.66)	50(20.08)	10(20)	1.44(0.47–4.41)	41(19.71)	15(12.82)	0.60(0.26–1.39)
Q3(53.66–61.74)	51(20.48)	16(32)	2.42(0.82–7.10)	43(20.67)	29(24.79)	1.20(0.54–2.68)
Q4(61.74–73.34)	49(19.68)	10(20)	1.13(0.35–3.68)	42(20.19)	28(23.93)	1.44(0.63–3.32)
Q5(>73.34)	50(20.08)	6(12)	0.70(0.17–2.90)	40(19.23)	19(16.24)	1.21(0.43–3.40)
*p* for linear trend ^6^			0.237			0.358
*p* for continuous ^7^			0.829			0.226
Dietary GL ^5^						
Q1(<139.44)	50(20.08)	7(14)	1 (ref.)	42(20.19)	12(10.26)	1 (ref.)
Q2(139.44–168.43)	50(20.08)	13(26)	2.42(0.76–7.65)	42(20.19)	20(17.09)	1.66(0.69–3.98)
Q3(168.43–190.88)	49(19.68)	10(20)	1.37(0.41–4.66)	42(20.19)	16(13.68)	1.16(0.47–2.92)
Q4(190.88–215.37)	50(20.08)	8(16)	1.39(0.38–5.02)	42(20.19)	16(13.68)	1.05(0.40–2.74)
Q5(>215.37)	50(20.08)	12(24)	2.39(0.65–8.78)	40(19.23)	53(45.3)	2.61(1.09–6.24)
*p* for linear trend ^6^			0.491			0.087
*p* for continuous ^7^			0.442			0.074

^1^ CINs, cervical intraepithelial neoplasia; GI, glycemic index., GL, glycemic load; HPV, human papilloma virus. ^2^ HPV was detected using a Hybrid Capture II assay for detection of all HPV types. ^3^ This indicates the number of participants and their percentage of the total. ^4^ Adjusted for age (years) and energy intake (kcal/day) as continuous variables and for marital status (single, married, widowed/divorced/separated), educational status (≤high school graduate, ≥college), physical activity (sufficient, insufficient), smoking status (non-smoker, past-smoker, smoker), history of pregnancy, oral contraceptive use (non-user, past-/current-), and family history of cervical cancer, and hospitals (H01, H03, Ho4, H05, H06, H07, H12, NCC) as categorical variables. ^5^ Dietary GI and GL, and carbohydrate were each divided into five groups based on the normal group. Q1 and Q5 were the lowest and highest quintile groups, respectively. ^6^
*p* value was calculated for the linear trend of multivariate odds ratio. ^7^
*p* value was calculated by taking dietary GI and GL as continuous variables. Ref, standard ie. if Q1 is 1, others increases risk on odds ratio.

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
