# Peer review of "Associations of Dietary Glycemic Index, Glycemic Load and Carbohydrate with the Risk of Cervical Intraepithelial Neoplasia and Cervical Cancer: A Case-Control Study"

_nutrients, 2020, doi:10.3390/nu12123742_

Round 1

Reviewer 1 Report

The present study revealed the association of dietary GI and GL with the risk of CIN1 especially among women with BMI <23 and premenopausal.

This paper is well described as a case-control study paper in nutrition epidemiology.

HPV infection is a major risk factor for cervical cancer. I think it is better to include HPV infection as a confounder in the model of multivariate analysis. Or, I think it is better to confirm whether there is an effect of HPV infection by stratified analysis.

Does the dietary survey examine the dietary status at the time of the survey? In that case, do you assume that the dietary habits of the subjects do not change significantly over time?

How different is GI and GL between the average Korean diet and the Western diet?

In the present study, an association with CIN1 was observed, but no association with CIN2/3 and cervical cancer were observed. How do you interpret this result?

Author Response

Reviewer #1

Comments and Suggestions for Authors

First we would like to thank you for your valuable review and suggestions

The present study revealed the association of dietary GI and GL with the risk of CIN1 especially among women with BMI <23 and premenopausal.

This paper is well described as a case-control study paper in nutrition epidemiology.

Suggestions

HPV infection is a major risk factor for cervical cancer. I think it is better to include HPV infection as a confounder in the model of multivariate analysis. Or, I think it is better to confirm whether there is an effect of HPV infection by stratified analysis.

  • As per your suggestions, we have done stratified analysis by HPV infection and results were reported in the manuscript. (Lines – 197-202)

Does the dietary survey examine the dietary status at the time of the survey? In that case, do you assume that the dietary habits of the subjects do not change significantly over time?

  • Yes, the dietary survey examine the dietary status over the course of a year preceding enrollment, and in all epidemiological studies the usual dietary habits were collected by this method. (Line – 117)

How different is GI and GL between the average Korean diet and the Western diet?

  • The difference between average Korean diet and Western diet is explained in the discussion part of the manuscript.

In the present study, an association with CIN1 was observed, but no association with CIN2/3 and cervical cancer were observed. How do you interpret this result?

  • The clarification of the above statement is properly explained in the discussion part of the manuscript.

Reviewer 2 Report

  1. I disagree that this manuscript is a case-control study. Because the authors needed to select the cases first, then selected the controls base on cases in a case-control study. Please clarify the study design.
  2. Because the authors found “Dietary GI and GL were not associated with CIN2/3 and cervical cancer”, I think “the associations of dietary GI and GL with increased risk of CIN1” would be “by chance” or some confounding factors didn’t adjust, ex: multi-center or others.
  3. In addition, because of “Dietary GI and GL were not associated with CIN2/3 and cervical cancer”, I think the authors should not conclude that ”our findings may have important implications for reducing cervical cancer risk” in spite of the associations of dietary GI and GL with increased risk of CIN 1.
  4. Because of “The study subjects aged 18-65 years were screened from the gynecologic centers of eight university hospitals in Korea”, different hospitals should be adjusted in multivariate logistic regression model.
  5. line 123, …..“Tthe" histological, “Tthe" was typo.

Author Response

Reviewer #2

Comments and Suggestions for Authors
We would like to thank you for your valuable review and suggestions

I disagree that this manuscript is a case-control study. Because the authors needed to select the cases first, then selected the controls base on cases in a case-control study. Please clarify the study design.

  • The current study is a case-control study including 670 cases and 670 controls. Details of the research design is explained in the manuscript. (Lines – 81-92)

Because the authors found “Dietary GI and GL were not associated with CIN2/3 and cervical cancer”, I think “the associations of dietary GI and GL with increased risk of CIN1” would be “by chance” or some confounding factors didn’t adjust, ex: multi-center or others.

  • As per your suggestions, additional factors are added and results were reported. (Lines – 175-202)

In addition, because of “Dietary GI and GL were not associated with CIN2/3 and cervical cancer”, I think the authors should not conclude that “our findings may have important implications for reducing cervical cancer risk” in spite of the associations of dietary GI and GL with increased risk of CIN 1.

  • The conclusion part of the manuscript is corrected based on our research findings.

Because of “The study subjects aged 18-65 years were screened from the gynecologic centers of eight university hospitals in Korea”, different hospitals should be adjusted in multivariate logistic regression model

  • Different hospitals were adjusted in multivariate logistic regression model and results were reported in the manuscript. (Lines – 175-202)

line 123, …..“Tthe" histological, “Tthe" was typo.

  • The error was corrected. (Line – 135)

Round 2

Reviewer 2 Report

All comments have addressed. I have no more questions.